# Egress-enhancing mutation reveals the inefficiency of non-enveloped virus cell exit

**Valerie J. Rodriguez-Irizarry, Robert W. Maples, Julie K. Pfeiffer** [ID] *

Department of Microbiology, University of Texas Southwestern Medical Center, Dallas, Texas, United States of America,

* Julie.Pfeiffer@UTSouthwestern.edu

## Abstract

Viruses encounter a range of selective pressures, but inefficiencies during replication can be masked. To uncover factors that limit viral replication, we used forward genetics to enrich for a murine norovirus (MNV) mutant with faster replication. We sequentially harvested the earliest progeny in cultured cells and identified a single amino acid change in the viral NS3 protein, K40R, that was sufficient to enhance replication speed. We found that the NS3-K40R virus induced earlier cell death and viral egress compared with wild-type virus. Mechanistically, NS3-K40R protein disrupted membranes more efficiently than wild-type NS3 protein, potentially contributing to increased mitochondrial dysfunction and cell death. Immunodeficient mice infected with NS3-K40R virus had increased titers, suggesting that increasing egress did not reduce fitness *in vivo*. Overall, by using a forward genetic approach, we identified a previously unknown inefficiency in norovirus egress and provide new insights into selective pressures that influence viral replication and evolution.

## Introduction

Viral replication is a complex process and exposure to selective pressures can drive viral adaptation [1–3]. In particular, RNA viruses are highly adaptable due to the high mutation rates of RNA-dependent RNA polymerases, which lack proofreading [2,4–7]. Adaptive mutations can enhance viral replication under various selective pressures, including neutralizing antibodies [8], replication within animals [9], and heat exposure [10]. Forward genetic screening has revealed viral adaptive changes and illuminated unexpected replication defects. For example, in a previous study, we demonstrated that a mutation in the capsid of coxsackievirus B3 increased replication in cultured cells by destabilizing the virion and facilitating more efficient viral RNA (vRNA) release during uncoating [11]. Therefore, viral adaptive mutations can reveal new insights into unanticipated facets of viral replication.

Murine norovirus (MNV), a nonenveloped positive single-stranded RNA virus from the *Caliciviridae* family, serves as a model enteric virus [12,13]. The MNV genome is

provided the original author and source are credited.

**Data availability statement:** All relevant data are within the paper and its Supporting information files

**Funding:** National Institutes of Health grants R37 AI074668 to J.K.P., T32 AI007520 to V.J.R.I, and T32AI005284 to R.W.M. The funders did not play any role in the study design, data collection and analysis, decision to publish, or preparation of the manuscript.

**Competing interests:** The authors have declared that no competing interests exist.

**Abbreviations:** DMEM, Dulbecco's Modified Eagle Medium; FBS, fetal bovine serum; LOD, limit of detection; MBP, maltose-binding protein; MLKL, mixed lineage kinase domain-like; MNV, murine norovirus; ORFs, open reading frames; ROS, reactive oxygen species; SEM, standard error of the mean; vRNA, viral RNA.

approximately 7.4 kb and encodes 3–4 open reading frames (ORFs) [12,14]. Translation of ORF1 produces a polyprotein that is cleaved by the viral protease to generate the mature non-structural proteins NS1–7 [14]. The vRNA replication complex has been linked to vesicular clusters [15,16] and includes the RNA-dependent RNA polymerase (RdRp) (NS7), VPg (NS5), and nucleoside triphosphatase (NTPase)/RNA helicase (NS3) [17]. MNV lab strains share 87% or greater genetic identity [18] and can cause either acute (MNV1 CW3) or persistent infections (MNV3, MNV CR6, MNV CR3) in mice [19,20]. In cell culture, MNV replicates efficiently in dendritic and macrophage cell lines [21], and many groups use immortalized macrophage cell lines, such as BV2 and RAW264.7, for MNV studies. Replication kinetics can vary, but capsid proteins are detectable as early as 6 h post-infection, and progeny virions are typically observed 9–12 h after infection [21]. Therefore, the 12–24 h timeframe is frequently used for *in vitro* infection experiments.

NS3 is a multifunctional protein essential at different stages of replication. The NS3 C-terminal domain contains the evolutionarily conserved viral helicase and NTPase, both of which are required for viral genome replication [22,23]. Caliciviruses induce cell death [24–30], and previous studies have implicated NS3 in this process [31,32]. Recently, it was demonstrated that the N-terminal region of NS3 promotes viral egress [31,32]. Wang and colleagues showed that NS3 interacts with mitochondrial lipids, including cardiolipin, and disrupts cardiolipin-containing liposomes, leading to mitochondrial dysfunction and subsequent cell death [31,33]. Notably, in the absence of this NS3-mediated cell death function, infected cells accumulate large numbers of progeny viruses that fail to exit and spread. However, the large 158 amino acid domain deletion in NS3 in that study makes it difficult to disentangle effects on egress from other facets of replication.

In this study, we selected for MNV mutants with enhanced replication speed by sequentially harvesting the earliest progeny, at 6 h post-infection, in cultured cells. We found that a single amino acid change in the viral NS3 protein was sufficient to enhance replication. The NS3-K40R mutant induced earlier cell death and viral egress compared to wild-type MNV. Furthermore, the NS3-K40R mutant virus had increased titers in immunodeficient mice. Our findings suggest that viruses can adapt to selective pressures to accelerate replication, and forward genetic approaches are valuable tools for uncovering unanticipated factors that influence viral replication.

## Results

### Forward genetic screen for fast-replicating MNV mutants

To determine whether MNV could adapt to replicate more rapidly, we performed serial passage of MNV3 in BV2 cells, harvesting the early progeny. MNV progeny can be detected at high titers after 12 h post-infection [21,34]. For serial passage, we infected BV2 cells with MNV3 at an MOI 0.1 and we collected combined cells and culture supernatants for each sample at two different time points: 6 h post-infection to harvest the earliest progeny, or 24 h post-infection to harvest progeny from a complete replication cycle. The low-yield virus collected at 6 h was amplified in BV2 cells

overnight to achieve higher titers. Samples were then freeze-thawed to release intracellular virus, and half of the sample was used for a new passage. This process was repeated for 11 passages (Fig 1A). After 11 passages, we performed a viral infection assay for 6 h at an MOI 0.1 to determine the phenotype of the passaged virus compared to the input stock. Samples collected at 6 h post-infection (Passage 11-6 h) produced higher yields than the input stock virus (Passage 0) or samples collected at 24 h post-infection (Passage 11-24 h) (Fig 1B). We also performed a single-cycle growth curve to determine replication kinetics and observed higher viral titers from Passage 11-6 h compared to Passage 0 at early time points of infection (Fig 1C). For example, passage 11-6 h virus had 42-fold higher titers at 8 h post-infection compared with Passage 0 virus. These data suggest that the passaged virus (Passage 11-6 h) produced progeny viruses earlier than Passage 0 virus.

## A single mutation, NS3-K40R, is sufficient for the fast-replication phenotype

We performed consensus sequencing to determine whether mutations contributed to the increased replication kinetics of the Passage 11-6 h virus. Viral genomes from Passage 11-6 h and Passage 11-24 h were amplified using RT-PCR and products were sequenced. We identified a total of three mutations in Passage 11-6 h virus, at nucleotides 1147, 2047, and 7022 (Fig 2A). The mutation at nucleotide 2047 resulted in an amino acid substitution from isoleucine to threonine (I340T) in non-structural protein 3 (NS3), which was present in both Passage 11-6 h and Passage 11-24 h. Given that I340T

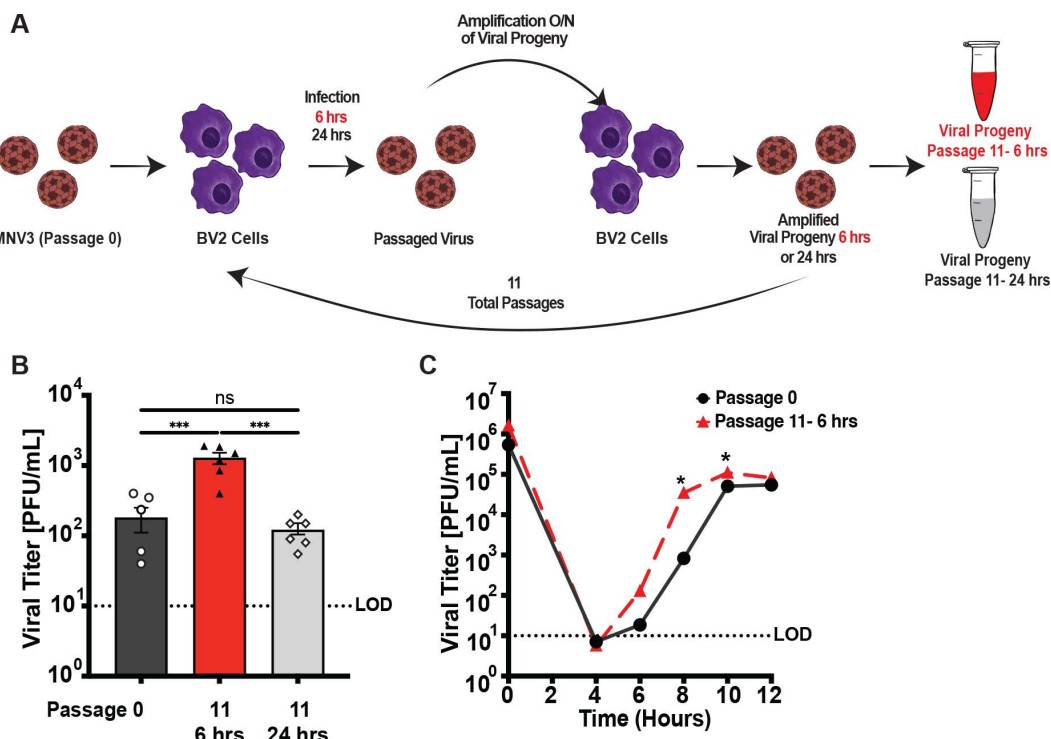

**Fig 1. Selection to enrich for fast-replicating MNV mutants. (A)** Schematic showing serial passage strategy. Initial MNV3 stock (P0) was used to infect BV2 cells, and progeny were harvested at 6 h or 24 h post-infection. Low titer viral progeny harvested at 6 h were amplified overnight in BV2 cells to facilitate subsequent passages. Half of the volume of each sample was used to initiate a new passage, and the cycle was repeated 11 times. **(B)** Viral infection assay comparing the initial stock virus (Passage 0) and viruses passaged 11 times, collecting at 6 h vs. 24 h. BV2 cells were infected at an MOI 0.1 for 6 h. Cells and supernatant were pooled for each sample, and viral titers were quantified via plaque assay. Data are mean ± SEM ($n = 6, \geq 3$ independent experiments). *$P < 0.05$, one-way ANOVA. **(C)** Single-cycle growth curve of Passage 0 and Passage 11-6 h. BV2 cells were infected at MOI 0.1 and cells and supernatant were collected and viral titer determined via plaque assay. Data are mean ± SEM ($n = 7, \geq 4$ independent experiments). *$P < 0.05$, unpaired $t$ test. LOD, limit of detection. The data underlying this Fig can be found in S1 Data file, Tabs 1–2.

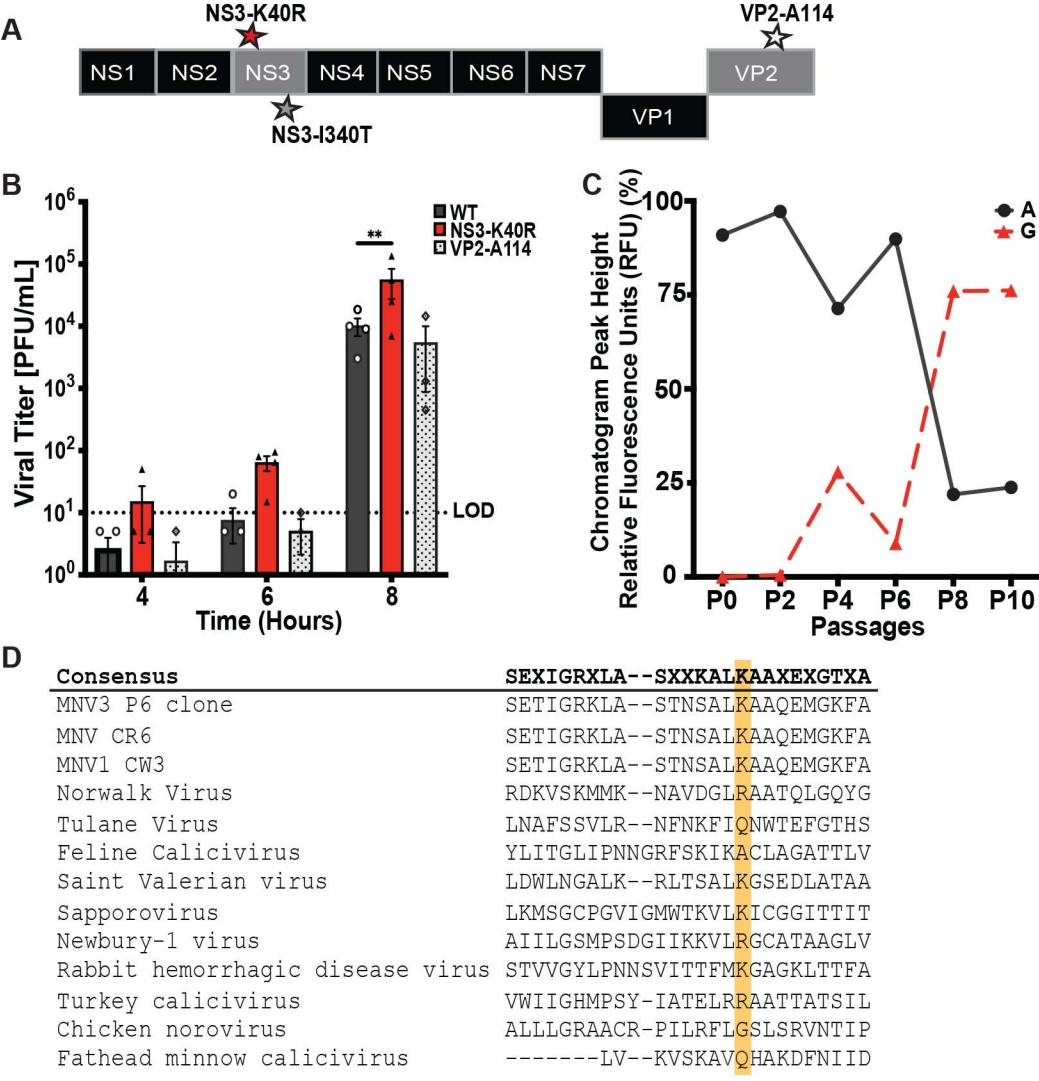

**Fig 2. NS3-K40R is sufficient for the faster MNV3 replication phenotype. (A)** Schematic of MNV3 genome with mutations found after 11 passages. Consensus sequence analysis revealed three mutations in viral progeny from passage 11-6 h stocks: Two amino acid substitutions in NS3, K40R, and I340T, and a silent mutation in the minor capsid protein VP2. Passage 11-24 h stocks also contained the NS3-I340T mutation. **(B)** Viral infection assay of MNV3 WT, NS3-K40R, and VP2-114. Infectious clone-derived viruses bearing NS3-K40R, the VP2-114 silent mutation, or no engineered mutations (WT) were used to infect BV2 cells at an MOI 0.1. Cells and supernatant were collected, and virus was quantified via plaque assay. Data are presented as mean ± SEM ($n = 4$; ≥ 4 independent experiments). *$P < 0.05$, two-way ANOVA. **(C)** Emergence of adenine to guanine (AAG to AGG) mutation at position 1147 (NS3 amino acid 40) during viral passaging. Peak heights were determined from chromatograms of the consensus sequence across passages. The percentages of peak heights were calculated based on the relative abundance of all four nucleotides at position 1147. **(D)** Conservation of amino acids at position 25 to 49 of NS3 across selected caliciviruses. Viruses are arranged based on their percentage identity relative to MNV3. Amino acid sequence alignment was performed using MUSCLE. Lysine 40 (K40) of NS3 is marked with a yellow rectangle. LOD, limit of detection. The data underlying this figure can be found in S1 Data file, Tabs 3–4.

was present in both selective (Passage 11-6 h) and non-selective (Passage 11-24 h) conditions, it is possible that this mutation is due to drift or general cell culture adaptation. Indeed, we recently identified a mutation at this site, I340V, in MNV genomes following adaptation to replication in HeLa cells [35]. Both mutations at I340 appear to be associated with general cell culture adaptation. On the other hand, two mutations were unique to the viral progeny from Passage 11-6 h: a

missense mutation nucleotide 1147 that resulted in substitution from lysine to arginine (K40R) in NS3 and a silent mutation at nucleotide 7022 in the minor capsid protein VP2 (A114).

To determine if the NS3-K40R or VP2-A114 mutations were sufficient to increase replication kinetics, we cloned them into an MNV3 infectious clone plasmid using site-directed mutagenesis and generated virus stocks. We infected BV2 cells at MOI 0.1, collected combined cells and supernatant for each sample at 4, 6, and 8 h post-infection, and viral titer was determined via plaque assay. NS3-K40R showed higher viral titers, most notably at 8 h post-infection, with ~5-fold higher titers than cells infected with WT (Fig 2B). The titers from VP2 A114 were similar to WT. These data suggest that NS3-K40R was sufficient to confer increased replication kinetics. To assess when the NS3-K40R arose during passaging, we performed consensus sequence analysis for progeny from previous passages, focusing on the NS3 N-terminal region. Mutation of adenine to guanine (A<u>A</u>G to A<u>G</u>G), which confers the NS3-K40R substitution, was first observed at Passage 4, and predominated the population by Passage 8 (Fig 2C).

NS3 is a multifunctional protein with enzymatic activity in the viral replication complex [22,23] and a role in cell death and viral egress [31,32]. MNV NS3 protein has three domains: the N-terminal domain (amino acids 1–158), the core domain (amino acids 158–289), and the C-terminal domain (amino acids 290–364) [22,31]. Specifically, the NS3-K40R mutation is in the N-terminal domain of NS3. Lysine at position 40 is conserved among MNVs (Fig 2D, highlighted in yellow); however, it is less conserved among other members of the *Caliciviridae* family. For example, human norovirus has arginine and Tulane virus has glutamine at this position (Fig 2D). These data suggest that the impact of this region of NS3 may vary across the *Caliciviridae* family.

## NS3-K40R does not alter RNA synthesis or NTPase activity

To determine whether NS3-K40R enhances replication kinetics through effects on RNA replication and/or virion production, we quantified vRNA accumulation in infected cells and the ratio of vRNA to PFU. First, we quantified viral genome copy numbers early in infection. BV2 cells were infected at an MOI 2, harvested at 2, 4, and 6 h post-infection, and vRNA was quantified by quantitative RT-PCR (qRT-PCR). No differences in genome copies were observed between NS3-K40R and WT virus (Fig 3A). Second, we compared the replication capacity of WT and NS3-K40R by calculating the vRNA:PFU ratio. BV2 cells were infected at MOI 2, and cells were collected at 10 h post-infection. Half of the cell volume was used for qRT-PCR to quantify genome copies as described above, while the other half was used for plaque assays to determine viral titers. We then calculated the vRNA:PFU ratio by dividing the number of genome copies by the viral titer. No significant differences in the vRNA:PFU ratio were observed (Fig 3B), indicating that NS3-K40R does not alter vRNA or virion production at 10 h post-infection.

Next, to determine whether NS3-K40R alters NS3 enzymatic functions as an NTPase, we purified NS3 proteins and measured ATP hydrolysis via release of inorganic phosphate [23]. We expressed NS3 as maltose-binding protein (MBP) fusion proteins and purified full-length His6-MBP-NS3 WT and K40R using affinity chromatography. To ensure equal protein concentrations for the NTPase assay, purified proteins, and an MBP control were analyzed by SDS-PAGE and band intensities were quantified (Fig 3C). Then, WT or NS3-K40R proteins were incubated with ATP for 30 min at 37 °C, and phosphate release was measured using a colorimetric assay. ATP was efficiently hydrolyzed by MBP-NS3, confirming its NTPase activity. However, no apparent differences in ATP hydrolysis were observed between WT and K40R after 30 min of reaction. As a control, MBP alone did not show detectable phosphate release (Fig 3D). Overall, these data demonstrate that NS3-K40R does not alter genome copy numbers during infection or ATP hydrolysis of purified protein, suggesting that the mutation does not enhance RNA synthesis.

## NS3-K40R enhances viral egress

Given that NS3-K40R did not alter RNA synthesis or vRNA:PFU ratio, we examined a later stage in the replication cycle, viral release/egress. As a non-enveloped virus, MNV progeny are fully assembled intracellularly prior to release. To

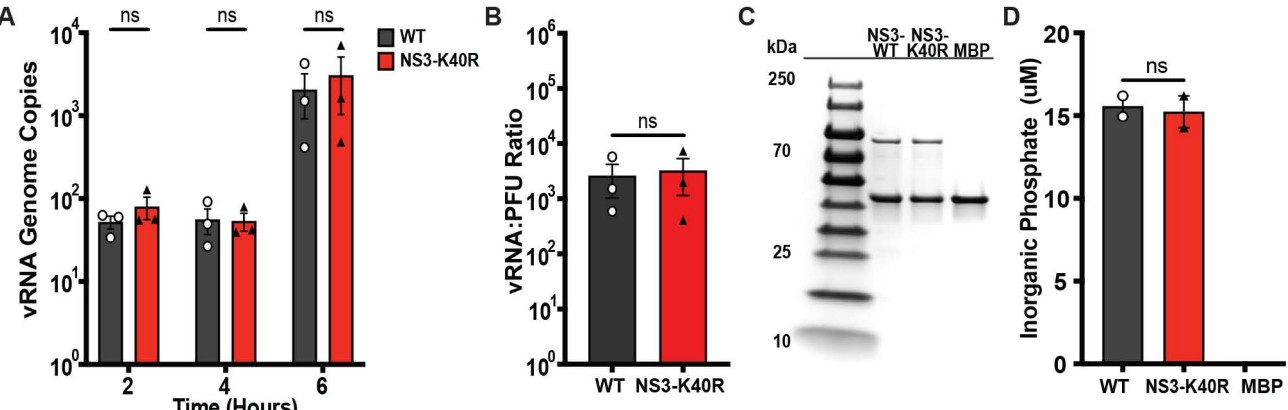

**Fig 3. NS3 K40R does not alter RNA synthesis or NTPase activity. (A)** Quantification of viral RNA during infection. BV2 cells were infected with WT or NS3-K40R at MOI 2, and intracellular viral RNA at indicated time points was harvested and quantified via RT-qPCR targeting MNV ORF1. Data are mean ± SEM ($n = 3$, ≥3 independent experiments). *$P < 0.05$, two-way ANOVA. **(B)** MNV3 WT and NS3-K40R have similar vRNA to PFU ratios. BV2 cells were infected and at 10 h post-infection cells were collected and split, with half used to determine titers by plaque assay and half used for RNA extraction and quantification by qRT-PCR. The vRNA:PFU ratio represents the viral genome copies (viral genomes/mL) divided by viral titer (PFU/mL). Data are mean ± SEM ($n = 3$, ≥3 independent experiments). *$P < 0.05$, unpaired $t$ test. **(C)** Purified MBP-NS3 proteins used for NTPase assay. MBP-NS3 fusion proteins or MBP protein alone were purified and analyzed by 4%–20% SDS-PAGE, followed by Coomassie Blue staining. The bands of proteins were quantified using ImageJ. **(D)** NTPase activity of purified NS3 proteins. NS3-MBP fusion proteins or MBP protein alone were incubated with 10 μM ATP for 30 min at 37 °C, and inorganic phosphate released from ATP hydrolysis was quantified via a colorimetric assay and its concentration was determined based on a standard curve. Data are mean ± SEM (Representative data from three independent protein preparations). *$P < 0.05$, ANOVA. The data underlying this figure can be found in S1 Data file, Tabs 5–7.

examine egress, BV2 cells were infected at an MOI 0.1 and viral progeny were collected at 4, 6, and 8 h post-infection from two distinct sites: cell-associated progeny within cells versus extracellular progeny in the media. Plaque assays revealed no differences in the viral titers of cell-associated virus between NS3-K40R and WT-infected cells, further supporting the idea that NS3-K40R does not alter RNA synthesis or earlier replication stages (Fig 4A). However, NS3-K40R virus had increased extracellular viral titers compared with WT, including a ~7-fold increase in viral yield at 8 h

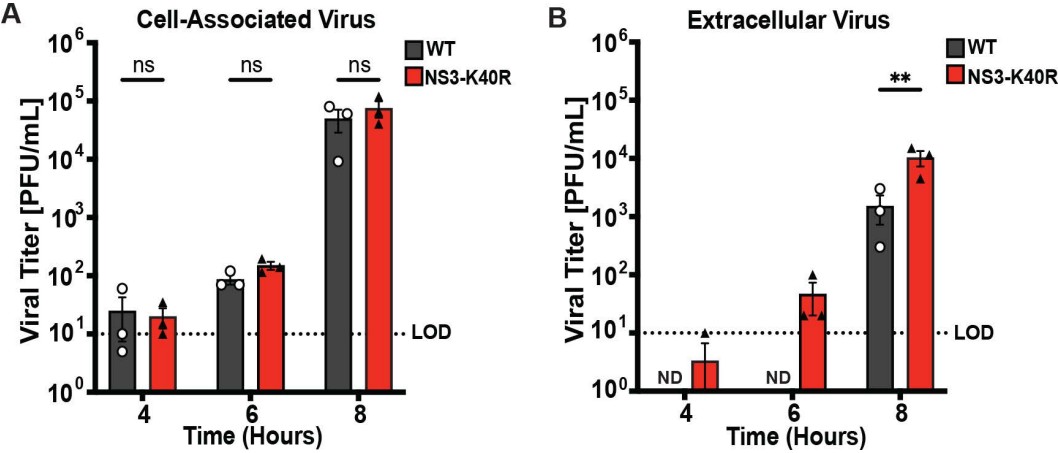

**Fig 4. NS3-K40R increases extracellular viral titers but not cell-associated viral titers.** BV2 cells were infected with MNV3 WT or NS3-K40R at MOI 0.1 for 4, 6, and 8 h post-infection, and progeny were harvested from **(A)** cells (representing virion particles within cells) or **(B)** culture supernatants (representing extracellular virus) followed by plaque assay quantification. Data are mean ± SEM ($n = 3$, ≥3 independent experiments). *$P < 0.05$, two-way ANOVA. ND, not detected. LOD, limit of detection. The data underlying this figure can be found in S1 Data file, Tabs 8–9.

post-infection (Fig 4B). To gain finer insight into egress kinetics, we performed additional infection experiments where BV2 cells were infected at an MOI 0.1 and extracellular viral progeny in the media were collected every two hours over a 16-h period. We found that at early time points, between 6 and 10 h post-infection, the NS3-K40R had higher viral titers than WT (S1A Fig). However, at later time points, no significant differences were observed between WT and NS3-K40R. To determine the total viral yields for WT and NS3-K40R over the 16-h infection, we calculated the area under the curve and found no significant differences (S1B Fig). These data suggest that while NS3-K40R may confer an early advantage in viral release, the overall viral yield across the infection time course is similar for NS3-K40R and WT.

To determine whether effects extend to cell lines other than BV2 cells, we quantified extracellular viral yield of WT versus NS3-K40R in RAW264.7 macrophages. We found increased viral yield of NS3-K40R virus in the supernatant of RAW264.7 cells after 8 h post-infection (S2 Fig), suggesting that NS3-K40R facilitates earlier viral egress in multiple cell lines.

## NS3-K40R enhances cell death and membrane disruption

How does NS3-K40R induce earlier viral egress? Given that the K40R mutation is within the N-terminal domain of NS3 that induces cell death by targeting mitochondria [31], we examined cell death and production of mitochondrial reactive oxygen species (ROS) both late in infection and at early time points. First, to determine whether NS3-K40R affects cell viability late in infection, BV2 cells were infected with either WT or NS3-K40R virus, and cell viability was measured at 24 h post-infection using an ATP-based assay. At 24 h post-infection, cell viability was reduced for both WT and NS3-K40R compared with mock-infected cells. However, NS3-K40R virus induced greater cell death than WT virus (S3A Fig). Given that NS3-K40R promotes earlier viral egress at 6 and 8 h post-infection (Fig 4B), we next examined its effect on cell death at these early time points using SYTOX nucleic acid stain and flow cytometry. As expected, cell death was minimal this early in infection. However, NS3-K40R virus-induced increased cell death at 6 and 8 h post-infection compared with WT virus (Figs 5A, 5B, and S5A). Next, to assess mitochondrial damage, cells were infected as above but incubated with MitoSOX and quantified by flow cytometry to measure mitochondrial ROS. Mirroring the cell viability results, at 24 h post-infection, both WT and NS3-K40R infected cells had higher mitochondrial superoxide than mock-infected cells, but NS3-K40R virus-induced higher levels than WT virus (S3B Fig). At 6 and 8 h post-infection, mitochondrial superoxide levels were low, but NS3-K40R virus-induced increased mitochondrial superoxide compared with WT virus (Figs 5C, 5D, and S5B). These findings suggest that NS3-K40R promotes both cell death and mitochondrial disruption earlier in infection compared to WT virus.

NS3 protein binds cardiolipin [31], a phospholipid primarily localized in the inner membrane of mitochondria but that is translocated to the outer membrane during cellular stress such as infection [36–38]. To investigate whether NS3-K40R induces cell death by disrupting mitochondria, we tested its ability to disrupt cardiolipin liposomes. Due to potential confounding effects from the His6-MBP tag on NS3 proteins for these liposome-based experiments, we cleaved the tag, purified NS3 proteins via size-exclusion chromatography, and collected predicted monomers (~39 kDa) (S4A Fig). We then incubated varying concentrations of proteins with or without cardiolipin liposomes encapsulating terbium (III) cation ($Tb^{3+}$) and quantified membrane disruption through release of $Tb^{3+}$. As expected, there was minimal $Tb^{3+}$ release in control experiments using BSA with liposomes or NS3-K40R in the absence of liposomes (S4B Fig). At high NS3 protein concentration (0.5 µM), WT and K40R proteins showed no differences in their ability to disrupt cardiolipin liposomes (S4C Fig). However, at lower concentrations (0.05 and 0.005 µM), NS3-K40R was still able to disrupt cardiolipin liposomes, whereas NS3 WT was not (Fig 5E). Cardiolipin membrane disruption by NS3-WT required high protein concentrations (0.5 µM or 0.1 µM) (Fig 5F). These results suggest that NS3-K40R can disrupt cardiolipin membranes at protein concentrations 10–100-fold lower than WT NS3. Overall, these data indicate that the K40R mutation enhances the ability of NS3 to target cardiolipin-containing membranes, potentially contributing to mitochondrial dysfunction and cell death.

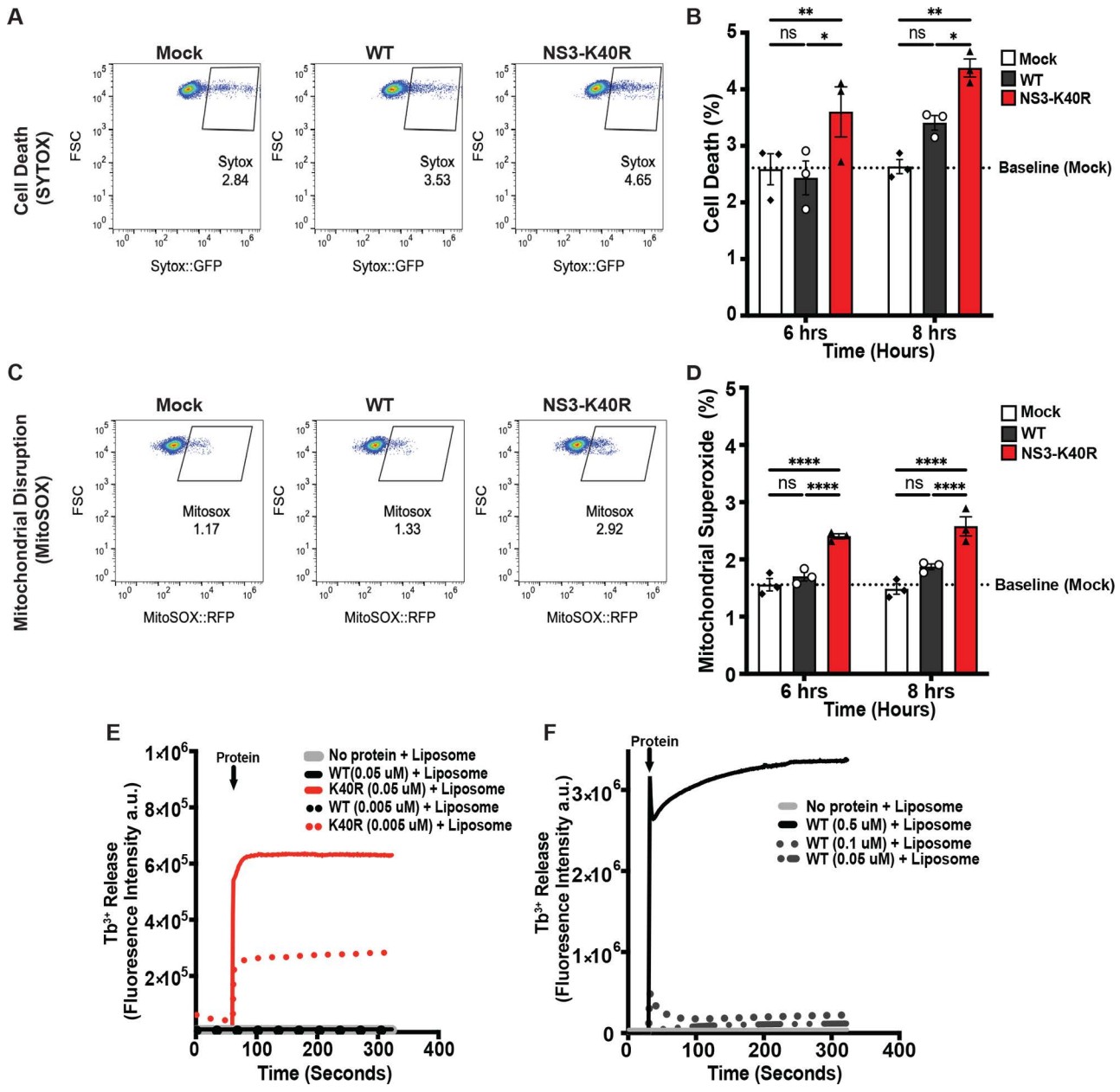

**Fig 5. NS3-K40R induces increased cell death, mitochondrial dysfunction, and membrane disruption. (A, B)** NS3-K40R increases cell death at early times post-infection. BV2 cells were infected with mock, WT or NS3-K40R at MOI 15 and at 6 or 8 h post-infection the number of dead cells was quantified by flow cytometry to detect SYTOX nucleic acid staining of non-viable cells. (A) Representative SYTOX flow cytometry data. (B) Compiled SYTOX data from multiple experiments. Values represent percentage of positive cells. Dashed line represents the baseline established from mock-treated samples. Data are mean±SEM ($n=3$, ≥3 independent experiments). *$P<0.05$, two-way ANOVA. **(C, D)** NS3-K40R increases mitochondrial superoxide at early times post-infection. BV2 cells were infected with mock, WT or NS3-K40R at MOI 15 and at 6 or 8 h post-infection mitochondrial superoxide was quantified by flow cytometry to detect MitoSOX staining. (C) Representative MitoSOX flow cytometry data. (D) Compiled MitoSOX data from multiple experiments. Values represent percentage of positive cells. Dashed line represents the baseline established from mock-treated samples. Data are mean±SEM ($n=3$, ≥3 independent experiments). *$P<0.05$, two-way ANOVA. **(E, F)** Cardiolipin liposome leakage assays with purified-tagless NS3 proteins. (E) Liposome leakage assay with low concentrations of NS3 proteins reveals increased activity of NS3-K40R compared with NS3-WT. Cardiolipin liposomes (0.1 mM; 80% phosphatidylcholine, 20% cardiolipin) were incubated with purified NS3-WT and NS3-K40R proteins at 0.05 µM and 0.005 µM and Tb$^{3+}$ fluorescence leakage was quantified upon binding to dipicolinic acid (DPA). Data are representative of two independent experiments with separate protein/liposome preparations. (F) NS3-WT protein induces liposome leakage at high protein concentrations. After the experiment in panel E, the assay was repeated using higher concentrations of NS3-WT protein (0.5, 0.1, and 0.05 µM). a.u., arbitrary units. The data underlying this figure can be found in S1 Data file, Tabs 10–13.

**NS3-K40R has increased replication in Stat1⁻/⁻ mice, but not in immune-competent mice**

To determine whether NS3-K40R alters MNV replication or pathogenesis *in vivo*, we perorally infected *Stat1⁻/⁻* mice with $10^6$ PFU of MNV3 WT or NS3-K40R. We chose to use immunodeficient *Stat1⁻/⁻* mice for these initial experiments because MNV can induce clinical signs and even death, whereas MNV infection is generally not severe in immunocompetent mice [39,40]. First, we monitored weight loss and clinical signs of infection. Mice were euthanized when they had lost 25% of their body weight or exhibited any critical clinical signs. In NS3-K40R-infected mice, clinical signs such as eye discharge appeared as early as 3 days post-infection. Both WT- and NS3-K40R-infected mice displayed similar clinical signs, including constipation, eye discharge, paralysis, and lethargy. Throughout the infection, NS3-K40R did not cause significant differences in weight loss or survival (Fig 6A and 6B). Second, we repeated the experiment in *Stat1⁻/⁻* mice as well as in immunocompetent mice but sacrificed mice at 24 h post-infection for titer analysis via plaque assays [41]. NS3-K40R-infected *Stat1⁻/⁻* mice had increased viral titers in the spleen, duodenum, ileum and colon (Fig 6C–6F). However, we did not observe differences in WT versus NS3-K40R viral titers from immunocompetent mice (Fig 6G and 6H). Overall, the increased titers of NS3-K40R in *Stat1⁻/⁻* mice suggest that this mutation enhances viral replication early in infection within an immunocompromised host, but this advantage is lost in mice with intact innate immune responses.

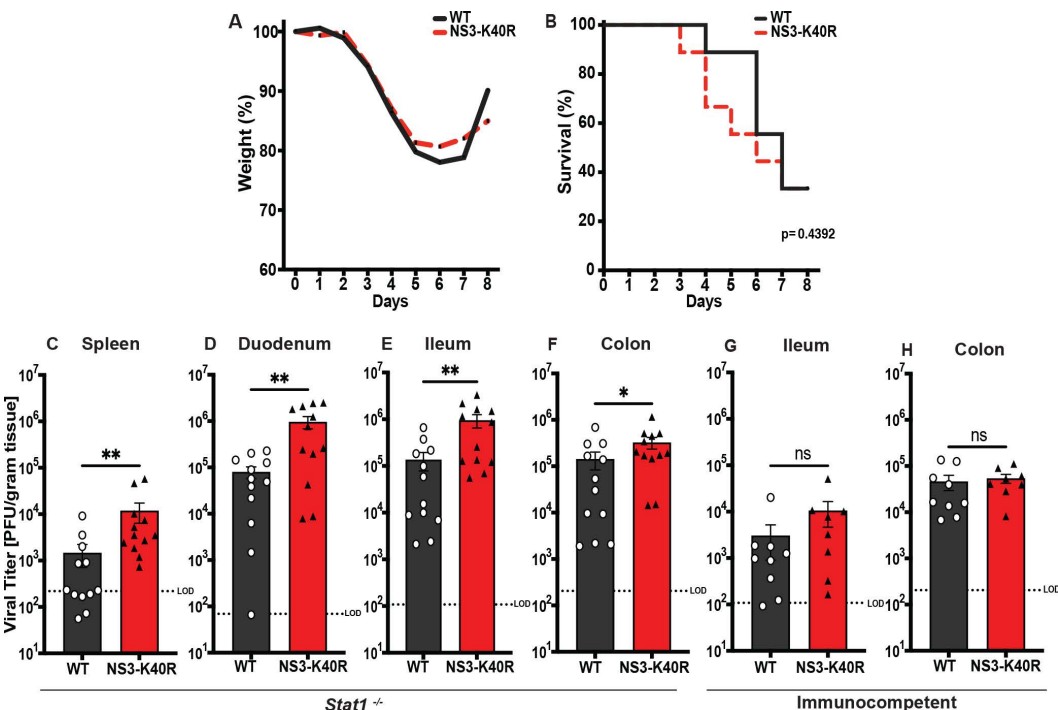

**Fig 6. NS3-K40R increases viral titers in *Stat1⁻/⁻* mice but has no effect in immunocompetent mice.** *Stat1⁻/⁻* mice were perorally infected with $10^6$ PFU of WT or NS3-K40R virus followed by assessment of **(A)** weight loss, **(B)** survival until 8 days post-infection. Data are mean ± SEM ($n = 8$, ≥ 2 independent experiments). *$P < 0.05$, Mantel-Cox test. Infections were repeated, but mice were sacrificed at 24 h post-infection for viral titer analysis via plaque assay for tissues including **(C)** spleen, **(D)** duodenum, **(E)** ileum, and **(F)** colon. We repeated infections in immunocompetent mice and sacrificed at 24 h post-infection for viral titer analysis via plaque assay for tissues including **(G)** ileum and **(H)** colon. Data are mean ± SEM ($n = 12$, ≥ 2 independent experiments). *$P < 0.05$, unpaired $t$ test. LOD, limit of detection. LOD was determined based on the average tissue weight. The data underlying this figure can be found in S1 Data file, Tabs 14–21.

## Discussion

While much is known about factors that influence viral replication, masked inefficiencies may exist that can be discovered using the power of forward genetic screens. Here, found that MNV egress can be enhanced by a mutation in NS3. This mutant virus had increased egress via enhanced cell death (Figs 4, 5A, and 5B). More specifically, the NS3-K40R protein disrupted membranes with mitochondrial lipids more efficiently than WT protein (Fig 5E and 5F). NS3-K40R virus also had increased titers in *Stat1⁻/⁻* mice (Fig 6C–6F) but was equivalent to WT virus in immune-competent mice. Overall, these results revealed a previously unknown inefficiency in calicivirus replication and revealed new insights into mechanisms of non-enveloped viral egress.

Although mutation of norovirus NS3 at position 40 has not been described previously, other mutations in NS3 have been characterized [22,34,35,42]. For example, targeted mutations in the predicted phosphate-binding loop (K168Q) reduced NS3's enzymatic activities, reducing inorganic phosphate release or unwinding activity [22,42]. Arias and colleagues identified rare variants containing NS3-T91A in mice persistently infected with MNV3 [34], and we previously identified NS3-I340V among a large number of mutations in MNV1 variants with increased replication in HeLa cells [35]. In this study, we identified an NS3 mutation with no apparent differences in enzymatic activity but instead it facilitated earlier cell death and viral egress.

Recent studies have suggested a role for the N-terminus of NS3 in cell death induction, and our results extend and elevate the importance of NS3-mediated egress *in vitro* and *in vivo* [31,32,43,44]. Aktepe and colleagues demonstrated that amino acids 67–100 of NS3 induce host translation shutoff and cell death [32]. Wang and colleagues demonstrated that N-terminal NS3 amino acids 1–158 were necessary and sufficient to induce mitochondrial dysfunction and cell death [31]. Viral genomes lacking the N-terminal domain of NS3 were capable of RNA replication and forming progeny virions, but lack of cell death and egress meant they were unable to spread, form plaques, or produce progeny in mice [31]. However, the large deletion in NS3 makes it difficult to differentiate effects on egress from other aspects of viral replication and fitness. Here, we used an unbiased approach to enrich for faster-replicating mutants and found a single conservative amino acid change in the N-terminal domain of NS3 increased egress by enhancing membrane damage, mitochondrial dysfunction, and cell death (Fig 5). Thus, in a system where any mutation in the genome that increases early progeny production could be selected, the fact that an egress-enhancing mutation emerged highlights the importance and relevance of NS3-mediated MNV egress. Overall, our study cements a key role for the NS3 N-terminal region, particularly position 40, in mitochondrial membrane disruption, cell death, and MNV egress.

How does a conservative amino acid change from lysine to arginine at position 40 of MNV NS3 alter mitochondrial membrane disruption, mitochondrial dysfunction, and cell death? Previous work suggests that the N-terminal domain of MNV NS3 contains a mitochondrial localization signal, and a four-helix bundle homologous to the membrane-disruption domain of the host mixed lineage kinase domain-like (MLKL) cell death protein [22,31,45,46]. For MLKL, phosphorylation triggers a conformational change that results in association with the plasma membrane, followed by membrane permeabilization via the four-helix bundle domain [46–48]. For NS3, Wang and colleagues suggest that the mitochondrial localization signal in amino acids 1–20 of NS3 directs the protein to mitochondria followed by membrane disruption via four-helix bundle domain in amino acids 25–158. Deletion of the N-terminal 20 amino acids of NS3 prevented MNV-mediated cell death and egress, thus blocking detectable viral spread *in vitro* and *in vivo* [31]. Although lysine to arginine is a conserved amino acid change, there is precedent for profound functional differences for proteins that bind and alter membranes. Previous studies have shown that replacing arginine with lysine reduces the membrane translocation efficiency of peptides like Penetratin [49]. In contrast, substituting lysine for arginine enhances cellular uptake of HIV Tat protein [50]. Notably, we found that NS3-K40R protein disrupts cardiolipin-containing membranes at concentrations 10–100-fold lower than wild-type NS3 protein (Fig 5E). Additionally, we routinely purified more NS3-K40R protein compared with NS3-WT protein, with ~3–6 times more NS3-K40R than WT in eight independent protein preparations (see S4A Fig for an example). The properties of arginine and its guanidinium group in NS3-K40R may influence protein stability and interactions with

membrane lipids, ultimately leading to mitochondrial disruption and cell death. Although the precise mechanism by which NS3-K40R disrupts mitochondria remains unknown, these results provide new insight into NS3-mediated viral egress.

Here we show that NS3-K40R MNV3 produces progeny faster *in vitro* and replicates to higher titers in *Stat1*<sup>−/−</sup> mice—so why does wild-type MNV3 contain lysine rather than arginine at position 40? In general, MNV strains contain lysine at this position (Fig 2D). However, a few caliciviruses have arginine at this position, including human norovirus (Norwalk strain), Newbury-1 virus, and Turkey calicivirus. The relative egress efficiencies of these other viruses are unknown, but it is possible that other amino acids in the N-terminal region of NS3 also impact cell death and egress. For example, early cell death and egress of progeny may reduce viral yield in the long run—a cell that dies early may contain fewer progeny viruses than a live cell that continues to support logarithmic viral replication for hours. Indeed, we found that over a full 16 h single cycle of infection, total yields of WT and NS3-K40R progeny were equivalent (S1 Fig), suggesting that WT virus "catches up". Another possibility is that early release of progeny induces early host immune responses that would then limit replication. Importantly, although NS3-K40R has higher titers than WT in *Stat1*<sup>−/−</sup> mice, this titer difference is lost immune-competent animals (Fig 6C–H). Therefore, it is possible that in immune-competent animals, STAT1-dependent innate responses are triggered sooner by early viral egress of NS3-K40R virions, thus limiting the advantage of NS3-K40R. Finally, a major determinant of viral fitness is interhost transmission. It is possible that variants with enhanced egress have reduced transmission. In the end, subtle fitness costs for NS3-K40R MNV3, or fitness costs not studied here may explain maintenance of lysine at this position. Overall, this study illuminates the importance of NS3-mediated egress of MNV and provides new insights into viral evolution and fitness.

## Materials and methods

### Ethics statement

All animals were handled according to the Guide for the Care of Laboratory Animals of the National Institutes of Health. All mouse studies were performed at University of Texas Southwestern Medical Center (Animal Welfare Assurance no. a3472-01) using protocols approved by the University of Texas Southwestern Medical Center Institutional Animal Care and Use Committee in a manner designed to minimize pain, and any animals that exhibited severe disease were euthanized immediately with isoflurane or $CO_2$.

### Cells and viruses

BV2 and RAW264.7 cells were maintained in Dulbecco's Modified Eagle Medium (DMEM) supplemented with 10% fetal bovine serum (FBS), 1% HEPES, and 1% penicillin-streptomycin. MNV3 P6 clone (Courtesy of Stephanie Karst Lab KC792553.1 [34]). Viruses were generated by transfecting HEK293T cells with infectious clone plasmids, followed by two rounds of amplification in BV2 cells to produce high-titer viral stocks. Stocks were stored at −80 °C.

### Viral passaging experiments

BV2 cells were seeded at $1.2 \times 10^6$ cells/well in 6-well plates and infected with MNV3 at an MOI 0.1. After 1 h at 37 °C, the supernatant was removed, cells were washed, and fresh media was added. Infection was continued for 6 or 24 h before scraping cells into the media to collect pooled intracellular and extracellular progeny. Samples were freeze-thawed three times to release intracellular virus, and half of the sample was used to initiate the next infection. Low yield 6 h passaged samples were used to infect new BV2 cells in overnight amplifications to facilitate the next passages. Passages 2-5 were performed at unknown MOI. Viral titers were determined by plaque assay after passage 5. Passaging continued with an initial MOI 0.1 for passage 6, followed by additional passages of unknown MOI. These forward genetic experiments with MNV were performed under BSL2 conditions, as approved by the UT Southwestern Institutional Biosafety Committee.

## Sequencing and cloning

We performed consensus sequencing on Passage 11 viral populations as previously described [35]. Briefly, Passage 11 viruses were used to infect BV2 cells, vRNA was isolated at 18 h post-infection using TRIzol reagent, and cDNA was generated by RT-PCR with SuperScript IV reverse transcriptase (ThermoFisher) using Primer X (Table 1). PCR products were made using primer sets from Table 1 and purified with the Nucleospin Gel and PCR Clean-up kit (Macherey-Nagel). Sequencing was performed by Azenta/Genewiz with the primers listed in Table 1. The sequences of the passaged viruses were compared with the sequence of initial stock (Passage 0) in SnapGene to identify mutations. We identified three mutations in Passage 11-6 h virus: A to G at nt 1147 (NS3-K40R), U to C at nt 2047 (NS3-I340T), and C to U at nt 7022 (a silent mutation in VP2 amino acid 114). Note that other minor variants may be present in P11-6 h samples that were not abundant enough to detect by consensus sequencing, but may contribute to phenotypes shown in Fig 1. The Passage 11-24 h samples also contained the NS3-I340T mutation, but not the NS3-K40R or silent mutation in VP2. It is possible that the NS3-I340T mutation, in combination with NS3-K40R, contributes to increased replication speed of the P11-6 h viral population (Fig 1C), but this was not evaluated here.

To determine at which passage lysine was substituted with arginine at NS3 amino acid 40, we analyzed the nucleotide sequences at position 1147. We analyzed the consensus sequence across all viral progeny passages at 6 h using Primer set II from Table 1. The peak heights of all four nucleotides (A, G, C, T) at position 1147 were determined from the sequencing chromatogram. The percentages presented in Fig 2C were calculated based on the relative abundance of each nucleotide.

Mutations unique to the Passage 11-6 h samples (NS3-K40R or VP2 silent mutation) were introduced into the MNV3 plasmid using site-directed mutagenesis (Genewiz/Azenta Life Sciences). Whole-plasmid sequencing (Plasmidsaurus) confirmed the presence of the mutations of interest and absence of other mutations. NS3-K40R and VP2-A114 viruses were generated from plasmids as described above. The presence of the mutations in the resulting viral stocks was sequence confirmed in RT-PCR products by Genewiz/Azenta Life Sciences.

## Phylogenetic analysis

We analyzed the conservation of amino acids at positions 25–49 of NS3 in relation to the MNV3 P6 clone. Sequence data were obtained from the NCBI database, with accession numbers and related information provided in Table 2. Representative sequences for caliciviruses were selected from the International Committee on Taxonomy of Viruses (ICTV) for each genus. Amino acid sequences were analyzed and aligned using MUSCLE, and the alignments were visualized with Geneious Prime.

**Table 1. Primers covering the MNV genome.**

|  | Forward (5′→3′) | Reverse (5′→3′) |
| --- | --- | --- |
| I | GT GAA ATG AGG ATG GCA ACG | G GGT CCA AAA GAT GTC AAA GA |
| II | G CTC AAC ATT CTC AAC ATC G | GAC CGC CTC CAG GTT GAC |
| III | GAC AGG ATT GAG AAC AAG GG | GTA ATC ATC AAT GGA GTA CTT G |
| IV | CAT GGA TAC ACC TAC CGT GA | CTG GAA CTC CAG AGC CTC AA |
| V | C ACC TGG GTT GTG ATT GGG | AT GTG GTA CCT GAA ATT GGC |
| VI | GCC AAC AAC ATG TAT GAG ATG | AGT CCT GTA ATA CTT TTC ACC A |
| VII | ACG CTG CAG GGC ATC TCC | CG ACC ATC CGG TAG ATG GT |
| VIII | T GGT CTC TGG CCG CCT TC | CTT CCC ACA GAG GCC AAT TG |
| IX | A CAC CGC TGA CGC CGC AG | AG CAG TAA GCA GAA ATC ATT TTC |
| X |  | TT TTA AAA TGC ATC TAA ATA CTA CT |

**Table 2. List of accession numbers from caliciviruses used for conservation analysis.**

| Calicivirus | General name used Fig 2D | Accession | Amino acid Accession no. |
|---|---|---|---|
| MNV1 CW3 | Murine norovirus | EF014462.1 | ABJ98943.1 |
| MNV3 P6 clone | Murine norovirus | KC792553.1 | AGO61997.1 |
| MNV CR6 | Murine norovirus | JQ237823.1 | AEY83582.1 |
| Norwalk virus | Human norovirus | M87661.2 | AAB50465.1 |
| Bavaria virus | Chicken norovirus | HQ010042 | ADN88287 |
| Lagovirus | Rabbit hemorrhagic disease virus | M67473 | AAA47285.1 |
| Minovirus | Fathead minnow calicivirus | KX371097 | AQM56929.1 |
| Nacovirus | Turkey calicivirus | JQ347522 | AFH89833.1 |
| Nebovirus | Newbury-1 virus | DQ013304 | AAY60849.1 |
| Recovirus | Tulane virus | EU391643 | ACB38131.1 |
| Sapovirus | Sapporovirus | HM002617 | ADG03646.1 |
| Valovirus | Saint Valerian virus | FJ355928 | ACQ44559.1 |
| Vesivirus | Feline calicivirus | M86379 | AAA79326.1 |

## Viral infection assays

Plaque assays were performed as described previously. Briefly, virus was diluted in PBS+ (PBS supplemented with 100 μg/mL $CaCl_2$ and $MgCl_2$) and incubated with BV2 cells for 20–30 min at 37 °C for attachment. Cells were then overlaid with 1.5% methylcellulose in MEM containing 10% FBS, 1% penicillin-streptomycin, and 1% HEPES. Overlays were removed after 72 h [35,51]. We set the limit of detection (LOD) at 2 plaques in undiluted samples (10 PFU); therefore, in some cases, values were below this detection limit. Samples with 1 or more plaques were included in the statistical analysis.

For infection experiments and single cycle replication experiments, $1.2 \times 10^6$ BV2 cells/well in 6-well plates were infected at MOI 0.1, and at 1 hr post-infection inoculum was aspirated, cells were washed, and fresh media was added. At each time point, cells and supernatant were pooled from each well and collected. Samples were subject to freeze-thaw cycles three times to release any intracellular virus, and viral titers were determined by plaque assay. For experiments in Fig 4 comparing cell-associated virus versus extracellular virus, infections were performed as above except that media were collected as the source of extracellular virus (with an additional spin to remove any floating cells) and remaining cells were washed prior to collection as the source of the cell-associated virus. For S1A and S1B Fig, the total area under the curve was analyzed using GraphPad Prism.

To quantify vRNA during infection we used qRT-PCR as previously described [51]. Briefly, $6 \times 10^6$ BV2 cells were infected with MNV3 WT and NS3-K40R virus at MOI 2 followed by removal of inoculum and washing at 1 hr post-infection. At 2, 4, and 6 hrs post-infection, cells were collected and intracellular RNA was isolated using TRIzol. cDNA was generated, and TaqMan quantitative PCR (qPCR) for MNV was performed in triplicate for each sample as described previously to quantify viral genomes [51]. The primers used for all samples are specified in Table 3. MNV3 plasmid that had been linearized with EcoRI was used as a standard.

For RNA:PFU ratio, plates were seeded with $4.4 \times 10^6$ BV2 cells and infected with WT or NS3-K40R at MOI 2. At 1 hr post-infection inoculum was aspirated, cells were washed, and fresh media was added. After 10 h post-infection, the cells

**Table 3. List of previously published primers used for qPCR [51].**

| Forward Primer (5′→3′) | GTGCGCAACACAGAGAAACG |
|---|---|
| Reverse Primer (5′→3′) | CGGGCTGAGCTTCCTGC |
| Probe | [6-FAM]-CTAGTGTCTCCTTTGGAGCACCTA-[BHQ1] |

were collected, and half were resuspended in 1 mL PBS+ and used for plaque assays on BV2 cells. The remaining cells were resuspended in TRIzol, and vRNA was isolated and quantified using qRT-PCR as described above. The RNA:PFU ratio was calculated by dividing the genome copy number by the titer for each sample.

### Protein purification

WT and NS3-K40R NS3 proteins were purified from *E. coli* using a pMAL-c6T-His6-MBP vector as previously described [31]. Briefly, protein expression was induced overnight at 20°C with 0.25 mM isopropyl β-D-1-thiogalactopyranoside after $A_{600}$ reached 0.8–0.9. The proteins were affinity-purified by Ni-NTA agarose (Qiagen). To remove heat-shock proteins we did an ATP wash that included incubation for <5 min with 20 mM Tris-HCl, 300 mM NaCl and 5 mM ATP. Bound proteins were washed three times and eluted with buffer containing 20 mM Tris-HCl (pH 8.0), 300 mM NaCl, 1 mM TCEP and Imidazole 250 mM. The proteins were subsequently affinity-purified with amylose resin (NEB) and eluted with buffer containing 20 mM Tris-HCl (pH 8.0), 300 mM NaCl, 1 mM TCEP and 10 mM maltose. After purification, proteins concentration was determined and normalized using 4%–20% SDS-PAGE with Coomassie Blue staining. BSA was used as the standard, and total protein intensity was quantified with ImageJ. For NTPase experiments, we used His6-MBP tagged versions of WT or K40R NS3 proteins. However, for liposome leakage assays, we used preparations of these proteins without tags. The His6-MBP tag was removed by overnight TEV digestion at 16 °C and further purified with Ni-NTA agarose. We performed dialysis to exchange media buffer A (20 mM HEPES (pH 7.5) and 150 mM NaCl). Then, TEV-cleaved NS3 proteins were purified by Superdex 200 size-exclusion chromatography column (GE Healthcare), and the predicted monomers (~39 kDa) were collected. Previous studies have demonstrated the early aggregation of NS3 during collection [23,31]. Therefore, for liposome leakage assays, we focused on investigating the effect of NS3 as a monomer (S4A Fig).

### NTPase assay.

NTPase activity of NS3 proteins was measured using Phosphate Assay Kit - PiColorLock (Abcam). His6-MBP-NS3-WT or His6-MBP-NS3-K40R proteins, or a MBP alone control, at 2 uM were incubated with 10 μM ATP for 30 min at 37°C, and inorganic phosphate released from ATP hydrolysis was quantified by measuring the absorbance using Biotek Synergy LX. A standard curve was used to determine the linear concentration of inorganic phosphate released during the reaction.

### Cell death and mitochondrial disruption assays

Cell death was determined by measuring the levels of the nucleic acid stain SYTOX (ThermoFisher) and mitochondrial ROS were determined by measuring the mitochondria-specific superoxide indicator MitoSOX Red (ThermoFisher). For both assays, 2 x 10^5 BV2 cells were infected with WT or NS3-K40R viruses at MOI 2 for 24 h experiments or MOI 15 for 6–8 h experiments, inocula were removed and cells washed at 1 hr post-infection, fresh media was added, and at 6, 8, or 24 h post-infection the relevant assay was performed. To measure cell death, BV2 cells were collected and stained with SYTOX for 15 min at RT. To measure mitochondrial ROS production, BV2 cells were collected and stained with the mitochondria-specific superoxide indicator MitoSOX Red for 30 min at 37°C. Cells were washed with phosphate-buffered saline (PBS) and fluorescence was quantified using a Stratedigm benchtop flow cytometer. The percentage of cells was determined using FlowJo software.

For cell viability assays, cells were infected as described above and number of live cells was determined by measuring ATP levels using the CellTiter-Glo Luminescent Cell Viability Assay (Promega) according to the manufacturer's instructions. ATP levels were measured via luminescence using a Synergy LX plate reader.

### Liposome preparation and liposome leakage assay

Cardiolipin liposomes were prepared as described by Wang and colleagues [31]. Briefly, lipids were obtained from Avanti Polar Lipids To prepare Tb^3+-encapsulated liposomes, lipids 80% POPC (1-palmitoyl-2-oleoyl-glycero-3-phosphocholine)

and 20% cardiolipin were mixed in a glass vial. Dry lipid films were obtained by evaporating under a stream of air and then hydrated at room temperature with vortex in 300 µl buffer B (20 mM HEPES (pH 7.5), 100 mM NaCl, 50 mM sodium citrate and 15 mM $TbCl_3$). Liposomes were generated by extrusion of the hydrated lipids through 100 nm polycarbonate filter (Whatman) 30 times using a Mini-Extruder device (Avanti Polar Lipids). The $Tb^{3+}$ ions outside the liposome were removed by washing with buffer A (20 mM HEPES (pH 7.5) and 150 mM NaCl) on a centrifugal column (Amico Ultra-4, 100 K MWCO, Millipore). The liposomes were resuspended in buffer A and were stored at 4 °C and used within 24 h.

For liposome leakage assays, 0.1 mM of $Tb^{3+}$-encapsulated liposomes were suspended in 250 µl buffer C (20 mM HEPES (pH 7.5) 150 mM NaCl and 50 µM of DPA) and NS3 or control proteins were added. Due to the sensitivity of these assays, NS3 proteins were used within 3 days of size-exclusion chromatography. Fluorescence of $Tb^{3+}$ was monitored over time on a PTI spectrofluorometer using $\lambda_{Ex} = 490$ nm and $\lambda_{Em} = 276$ nm. At the end of the incubation, 1% Triton X-100 was added to measure complete release of $Tb^{3+}$. Liposome preparations can vary leading to discrepancies in overall signal intensity. Therefore, comparisons were made only between assays using a given liposome preparation. Fig 5E and 5F shows analyses performed with a single, shared liposome batch, whereas the data in S4 Fig were generated using an independent liposome preparation.

## Mouse infections

A mix of male and female *Stat1*$^{-/-}$ C57BL/6 or wild-type C57BL/6 mice between 6–8 weeks of age were obtained from Jackson Laboratories (Jax#012606, Jax#000664) and were single-housed and left to acclimate for 5–7 days. Mice were perorally inoculated with 25 µl of $10^6$ PFU MNV3 WT or NS3-K40R in DMEM with 5% FBS. At 24 h post-infection, mice were euthanized, and tissues were collected. The tissues were homogenized by bead beating as previously described [35] and viral titers were determined via plaque assay using BV2 cells. We set the LOD at three plaques in undiluted samples and considered the average tissue weight to calculate the PFU/grams tissues; therefore, in some cases, values were below this detection limit. Samples with 1 or more plaques were included in the statistical analysis.

For survival curve and weight loss experiments we used *Stat1*$^{-/-}$ C57BL/6 mice that were bred and maintained in the UT Southwestern specific pathogen-free animal care facility. After peroral infection as described above the mice were observed and weighed for 8 days post-infection. Mice with clinical signs of disease (lost 25% body weight and/or were lethargic, had eye discharge, or paralysis) were euthanized by $CO_2$. The remaining mice were euthanized at the end of the study.

## Statistical analysis

All error bars represent the standard error of the mean (SEM). Specific statistical test used for each experiment is indicated in the figure legends. For all pairwise comparisons, a *P*-value < 0.05 was considered significant and represented with a single asterisk. GraphPad Prism was used to perform all statistical analyses.

## Supporting information

**S1 Fig. NS3-K40R and WT viruses generate similar total viral yields over a full cycle of replication. (A)** Single-cycle growth curve of WT and NS3-K40R extracellular viruses. BV2 cells were infected at MOI 0.1 and supernatant was collected and viral titer determined via plaque assay. Samples without detectable plaques are displayed as 1 (below the limit of detection, LOD) for visualization purposes. Data are mean ± SEM (*n* = 3, ≥ 3 independent experiments). *$P < 0.05$, unpaired *t* test. **(B)** Total viral load is similar between WT and NS3-K40R. The area under the curves in panel A are displayed. Data are mean ± SEM (*n* = 3, ≥ 3 independent experiments). The data underlying this figure can be found in S1 Data file, Tabs 22–23.
(TIF)

**S2 Fig. NS3-K40R increases viral titer in the supernatant of RAW cells at an early time point post-infection.** RAW264.7 cells were infected with WT and NS3-K40R at MOI 0.1 for 8 h. Extracellular virus was measured via plaque assay. Data are mean ± SEM ($n = 3$, ≥ 3 independent experiments). *$P < 0.05$, unpaired $t$ test. LOD, limit of detection. The data underlying this figure can be found in S1 Data file, Tab 24.
(TIF)

**S3 Fig. MNV3 NS3-K40R induces increased cell death and mitochondrial dysfunction late in infection. (A)** NS3-K40R increases cell death at 24 h post-infection. BV2 cells were infected with mock, WT or NS3-K40R at MOI 2 and at 24 h post-infection cell viability was quantified using ATP-based assay. Data are mean ± SEM ($n = 4$, ≥ 3 independent experiments). *$P < 0.05$, one-way ANOVA. **(B)** NS3-K40R increases mitochondrial superoxide at 24 h post-infection. BV2 cells were infected with mock, WT or NS3-K40R at MOI 2 and at 24 h post-infection mitochondrial superoxide was quantified by flow cytometry to detect MitoSOX staining. Data are mean ± SEM ($n = 3$, ≥ 3 independent experiments). *$P < 0.05$, one-way ANOVA. The data underlying this figure can be found in S1 Data file, Tabs 25–26.
(TIF)

**S4 Fig. NS3 protein preparation and controls for the liposome leakage assays. (A)** Purification of tagless NS3 proteins for liposome leakage assays. Purified His6-MBP-NS3 fusion proteins were treated with TEV protease to remove the His6-MBP tag. Following cleavage, the proteins were purified using Ni-NTA resin and subjected to size-exclusion chromatography. Absorbance peaks from size-exclusion chromatography are shown (NS3-WT in black and NS3-K40R in red) and the peak at approximately 16–17 mL was collected for use in liposome leakage assays. (B and C) Controls for the liposome leakage assay. **(B)** NS3-K40R was incubated in the absence of liposomes, or 0.5 µM BSA was incubated with cardiolipin liposomes (0.1 mM; 80% phosphatidylcholine and 20% cardiolipin) and at the end of the experiment detergent was added to disrupt liposomes. The leakage of Tb³⁺ fluorescence leakage was quantified upon binding to dipicolinic acid (DPA). Data are representative of two independent experiments. **(C)** Experiment was repeated as in B, but cardiolipin liposomes (0.1 mM; 80% phosphatidylcholine and 20% cardiolipin) were incubated with purified NS3-WT and NS3-K40R proteins at 0.5 µM or no protein, and leakage of Tb³⁺ fluorescence leakage was quantified upon binding to DPA. Data are representative of two independent experiments with separate protein/liposome preparations. a.u., arbitrary units. The data underlying this figure can be found in S1 Data file, Tabs 27–29.
(TIF)

**S5 Fig. Primary data for flow cytometry.** Representative flow cytometry plots showing the gating strategy used to measure **(A)** cell death and **(B)** mitochondrial disruption. BV2 cells were first gated based on forward and side scatter to exclude debris and select viable cell populations. Single cells were then identified using FSC-H vs. FSC-A gating. Within the single cell population, SYTOX-positive cells were gated to quantify cell death (A), or MitoSOX-positive cells were gated to assess mitochondrial reactive oxygen species production (B). This gating strategy was applied to the data shown in Fig 5A–5D.
(TIF)

**S1 Data. Primary data for all figures.**
(XLSX)

**S1 Raw Image. Uncropped gel for Fig 3C.** Purified MBP-NS3 proteins used for NTPase assay. MBP-NS3 fusion proteins or MBP protein alone were purified and analyzed by 4%–20% SDS-PAGE, followed by Coomassie Blue staining. The bands of proteins were quantified using ImageJ.
(TIF)

## Acknowledgments

We thank Tiffany Reese and Guoxun Wang for many helpful conversations about MNV egress, Kim Orth, Wei Peng, Katie Kang, David Heisler, Maarten De Jong for advice on protein purification and liposome assays, and Rob Orchard for advice on MNV experiments. We thank Andrea Erickson, Kim Orth, and Tiffany Reese for helpful comments on the manuscript.

## Author contributions

**Conceptualization:** Valerie J. Rodriguez-Irizarry, Julie K. Pfeiffer.

**Data curation:** Valerie J. Rodriguez-Irizarry.

**Formal analysis:** Valerie J. Rodriguez-Irizarry.

**Investigation:** Valerie J. Rodriguez-Irizarry, Robert W. Maples.

**Methodology:** Valerie J. Rodriguez-Irizarry, Robert W. Maples.

**Project administration:** Julie K. Pfeiffer.

**Supervision:** Julie K. Pfeiffer.

**Validation:** Valerie J. Rodriguez-Irizarry.

**Visualization:** Valerie J. Rodriguez-Irizarry.

**Writing – original draft:** Valerie J. Rodriguez-Irizarry.

**Writing – review & editing:** Valerie J. Rodriguez-Irizarry, Robert W. Maples, Julie K. Pfeiffer.

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
