## [Editor Report · Decision Letter 0]

Dear Dr Pfeiffer,

Thank you for submitting your manuscript entitled "Egress-enhancing mutation reveals inefficiency of non-enveloped virus cell exit" for consideration as a Research Article by PLOS Biology.

Your manuscript has now been evaluated by the PLOS Biology editorial staff, as well as by an academic editor with relevant expertise, and I am writing to let you know that we would like to send your submission out for external peer review.

Once your full submission is complete, your paper will undergo a series of checks in preparation for peer review. After your manuscript has passed the checks it will be sent out for review. To provide the metadata for your submission, please Login to Editorial Manager (https://www.editorialmanager.com/pbiology) within two working days, i.e. by Mar 12 2025 11:59PM.

Kind regards,

Melissa

Melissa Vazquez Hernandez, Ph.D.

Associate Editor

PLOS Biology

---

## [Decision Letter · Decision Letter 1]

Dear Dr Pfeiffer,

Thank you for your patience while your manuscript "Egress-enhancing mutation reveals inefficiency of non-enveloped virus cell exit" was peer-reviewed at PLOS Biology. It has now been evaluated by the PLOS Biology editors, an Academic Editor with relevant expertise, and by three independent reviewers.

In light of the reviews, which you will find at the end of this email, we would like to invite you to revise the work to thoroughly address the reviewers' reports. As you will see below, majority of reviewers are positive about the relevance and novelty of the study, yet some concerns have raised during revision. Reviewer 1 raises a key concern regarding the mechanism by which early cell death translates to virion production. Additionally, Reviewer 1 inquires about the evolutionary advantage of the mutation and whether there are differences in titers between the K40R virus and the WT virus in an in vivo infection model.

Reviewer 2 suggests that you investigate the I340T mutation further and recommends conducting a direct competition assay. Reviewer 3 notes a significant gap in the current data, specifically the lack of direct evidence showing that the NS3 K40 mutation accelerates cell death, and also raise questions about the evolutionary implications of this mutant.

IMPORTANT: After discussions with the Academic Editor and the reviewers, we think that while most of the concerns should be addressed, these can be tackled through in vitro experiments, and we will not require additional in vivo experimentation for further consideration.

Given the extent of revision needed, we cannot make a decision about publication until we have seen the revised manuscript and your response to the reviewers' comments. Your revised manuscript is likely to be sent for further evaluation by all or a subset of the reviewers.

**IMPORTANT - SUBMITTING YOUR REVISION**

*Re-submission Checklist*

*Published Peer Review*

*PLOS Data Policy*

*Blot and Gel Data Policy*

Sincerely,

Melissa

Melissa Vazquez Hernandez, Ph.D.

Associate Editor

PLOS Biology

REVIEWERS' COMMENTS:

Reviewer #1:

The studies by Rodriguez-Irizarry et al. isolate and characterize a novel adaptive mutation in murine norovirus that enables increased early replication. They convincingly prove that the functional adaptive mutation is within NS3 and characterize molecular functions of purified WT and mutant NS3. They propose that the mechanism of increased early replication is due to increased mitochondrial disruption, cell death, and virion release during early infection. The study is well conceived, and most data is presented clearly with thoughtful discussion. My main critique is related to mechanistic uncertainty about how early death would translate to increased virion production. Specific points of critique follow

Major points

1. The major point of confusion surrounds how early disruption of mitochondria would relate to more PFU produced. It doesn't make sense to me that earlier death and more extracellular PFU would result in greater total PFU, but should just shift the proportion of intracellular/extracellular. Does this imply that virion assembly is regulated by cell death? One way this may be regulated is through more rapid production of encapsidated virions. The authors could quantify VP1 by western blot to determine whether there is earlier protein production.

2. Related to the prior point, is there a difference in genome:PFU ratio? Figure 3A shows no difference in viral genomes at the same time points where there is a difference in PFU shown in prior figures. Presumably these genomes are exclusively from the cell associated fraction, but this should be clearly stated in the legend and/or methods. In addition to clarity on the source of genomes in Fig. 3A, it would be informative to quantify genome:PFU ratio from cell and supernatant for WT and mutant viruses to determine whether there is a qualitative difference in the PFU produced.

3. In Figure 4, there appears to be less PFU in the extracellular virus fraction from each virus strain than from the corresponding cell-associated fraction. This is puzzling because the earlier phenotype of the combined fractions in figure 1 would be expected to most reflect the majority of PFU present in the cell-associated fraction. Is this due to technical considerations related to tittering only a portion of the extracellular fraction or can this be explained in some other way?

4. Text describing figure 3C states that there is no difference in enzymatic activity of WT/K40R. However, the data seem insufficient to support this claim because Fig. 3C data show an extent of activity, not a rate. The claim in the text should be revised, or kinetics curves should be furnished.

5. Investigators comment in the text that an amino acid 1-20 deletion of NS3 abrogates mitochondrial localization. Does the K40R mutation still have an effect in the context of a 1-20aa deletion? This may be beyond the scope if the current study but would go some way to strengthening the mechanistic model of mitochondrial involvement.

6. Are titers of K40R virus different from WT during persistent infection of a non-immunodeficient mouse? This may be beyond the scope but would substantially increase the authors ability to discuss the evolutionary advantage or lack thereof.

Minor points

1. Please show representative sytox and mitoSOX flow cytometry data.

2. In figures 5A-B and S3B-C, the y-axis scales vary substantially. On the face of it, this does not present a problem, but the amount of release in S3C stimulated by 0.5 uM K40R is ~1/3 that shown to be stimulated by 0.05 uM K40R in 5C. Were these experiments scaled to one another to compare max signal for each replicate? If so, can the investigators show these data? If not, can the investigators comment on how to reconcile these variations in leakage magnitude?

3. Units for figures 3C, 5A, 5B are difficult to interpret. 3C appears to represent a concentration, of inorganic phosphate released from ATP hydrolysis, presumably calculated from a standard curve of absorbance-based measurements; would be better presented as raw absorbance with standard curve or as simply "free Pi", eg. 5A and B are described as being the difference between % of infected and mock cells in methods but as simply "normalized to uninfected" in figures.

4. Several cases of data below the LOD in titer assays - figures 6c, 4b, 2b, S1. How is LOD determined in these cases? Are statistics performed on values below the LOD?

Reviewer #2:

This paper is well written, with a straight forward experimental design. It seeks to identify how murine norovirus may adapt to selective pressure requiring faster infection, replication or production rates. The authors passage and collect virus at 6 hours versus 24 hours actross 11 passages. They identify two amino acid changes in NS3, one of which is unique to the 6 hour early collection timepoint.

The paper confirms that this mutation results in faster virus replication, virus production and egress, both in cell culture and in vivo.

I have no critical concerns wth the experiments as they are performed and am convinced by the data.

There are two points I wish to raise that could be considered for additional experimentation, without which the study seem incomplete. However, I don't think it it absolutely essential to perform. These studies would not negate the current results, but they would possibly provide a more complete picture.

One, is the fact that the I340T mutation was not studied, because it appeared in both 6 and 24

timepoints and thus considered to be cell adaptation unrelated to early passage. However, one could imagine that it still somehow confers this similar increase in replication and egress, and that it simply emerges at the later time-point. There is also question of how the double mutant would behave. The discussion does mention that they observed a mutation at this position in another cell line (this time it was I340V, but I do not believe the phenotype of this mutation was studied there either. Of course this would ask for more experimental work and might confuse the intended narrative, but could be considered by the authors. Perhaps they have already tried.

The other question is whether the authors could perform a direct competition assay in vivo (better) or in vitro, that might indicate that while this mutation does improve egress and virus production, it might not so easily outcompete wildtype virus, and thus, it only emerges when selective pressure leans heavily in its favor.

These can both be experimentally addressed, or discussed.

Reviewer #3:

This manuscript investigates the replication efficiency of murine norovirus (MNV), a non-enveloped virus, focusing on the viral egress stage. The central finding is the identification and characterization of a single amino acid mutation in the MNV non-structural protein 3 (NS3), specifically a lysine to arginine change at position 40 (K40R), which significantly enhances viral egress and overall replication speed. The study employs a forward genetic approach to uncover this previously masked inefficiency in the viral life cycle.

Rodriguez-Irizarry and colleagues used a forward genetic screen involving sequential harvesting of early viral progeny (at 6 hours post-infection) to enrich for faster-replicating MNV mutants in cultured BV2 cells.

After 11 passages, they identified a virus (Passage 11-6 hrs) with significantly higher yields at early time points compared to the wild-type virus (Passage 0). As shown in Figure 1B, for the comparison between Passage 0 and Passage 11-6 hrs at 6 hours post-infection. Figure 1C also demonstrates for higher viral titers of Passage 11-6 hrs at early time points in a single-cycle growth curve.

Sequencing of the fast-replicating virus revealed three mutations, one of which, a missense mutation resulting in the K40R substitution in the NS3 protein, was identified as crucial for the enhanced phenotype. Further experiments using infectious clones with the individual mutations confirmed that NS3-K40R was sufficient to confer the faster replication phenotype. Figure 2B shows that the "NS3-K40R" mutant displayed significantly higher viral yields compared to the wild-type (WT) virus.

The study found that the NS3-K40R mutant induced earlier cell death compared to the wild-type virus. Figure 5A shows a significant increase in Sytox-positive dead cells at 6 and 8 hours post-infection for the NS3-K40R virus. The enhanced cell death correlated with increased mitochondrial dysfunction, as evidenced by elevated levels of mitochondrial superoxide production at early time points in cells infected with the NS3-K40R mutant. Mechanistically, the NS3-K40R protein exhibited a greater ability to disrupt cardiolipin-containing membranes in vitro compared to the wild-type NS3 protein, particularly at lower protein concentrations (Figure 5C). This suggests that the mutation enhances the membrane-disrupting activity of NS3, potentially leading to mitochondrial damage and cell death.

The NS3-K40R mutation did not significantly alter viral RNA synthesis or the NTPase activity of the NS3 protein. Figure 3A shows no significant difference in viral RNA levels between WT and NS3-K40R at most time points. Figure 3C demonstrates that "no differences in ATP hydrolysis were observed between WT and K40R". This indicates that the enhanced replication is likely due to improved egress rather than increased RNA production.

Infection of Stat1-/- mice (immunocompromised mice lacking the type I interferon receptor signaling) with the NS3-K40R virus resulted in increased viral titers in various tissues (spleen, duodenum, ileum, and colon) at 24 hours post-infection (Figure 6 C-F, P<0.05 for several tissue comparisons). This suggests that the egress-enhancing mutation does not reduce viral fitness in vivo in the absence of a robust interferon response.

While the K40 residue in NS3 is generally conserved as lysine among murine noroviruses, some caliciviruses have arginine at this position. The study raises interesting questions about the selective pressures that maintain lysine at this position in MNV, suggesting potential fitness costs associated with enhanced egress that were not observed in this study (e.g., impact on host immune response or interhost transmission).

This study provides novel insights into the replication strategy of non-enveloped viruses by revealing a previously unrecognized inefficiency in the egress process of MNV. In addition, the identification and characterization of the NS3-K40R mutation highlight the crucial role of NS3 in viral egress and its link to host cell death and membrane disruption. The findings have implications for understanding the selective pressures balance that shape viral evolution and fitness.

While the effects are subtle, the data strongly support the thoughtful and well-written interpretation of the results. The study presents compelling findings, but one notable shortcoming is the lack of direct evidence demonstrating that the NS3 K40 mutant is faster specifically because it accelerates cell death. The observed effect on mitochondrial function remains correlational, leaving room for alternative explanations. For example, it's possible that the increased speed of the mutant could be a secondary consequence of a virus replication function that wasn't fully explored or considered within this study. Further analysis of this potential mechanism could strengthen the conclusions.

Additionally, the evolutionary implications of these findings are intriguing. If the NS3 K40 mutant competes with the wild type (WT) over multiple passages, it raises the question: which variant is ultimately fitter? This could be addressed with targeted experiments in cell culture, but ideally, it would also be tested in animal models to provide a more comprehensive understanding. The central question remains why this mutant, despite showing higher titers in both cell culture and animal studies, has not emerged as a dominant variant through evolutionary processes. Could there be unknown selective pressures or trade-offs that prevent the mutant from outcompeting the WT in natural settings? Exploring these aspects would provide valuable insights into viral evolution and the dynamics of fitness trade-offs in infection and transmission.

---

## [Editor Report · Decision Letter 2]

Dear Dr Pfeiffer,

Thank you for your patience while we considered your revised manuscript "Egress-enhancing mutation reveals inefficiency of non-enveloped virus cell exit" for publication as a Research Article at PLOS Biology. This revised version of your manuscript has been evaluated by the PLOS Biology editors and the Academic Editor.

Based on our Academic Editor's assessment of your revision, we are likely to accept this manuscript for publication, provided you satisfactorily address the remaining editorial points. Please also make sure to address the following data and other policy-related requests.

a) We routinely suggest changes to titles to ensure maximum accessibility for a broad, non-specialist readership, and to ensure they reflect the contents of the paper. In this case, we would suggest a minor edit to the title, as follows. Please ensure you change both the manuscript file and the online submission system, as they need to match for final acceptance:

"Egress-enhancing norovirus mutation reveals the inefficiency of non-enveloped virus cell exit"

Please supply the numerical values either in the a supplementary file or as a permanent DOI’d deposition for the following figures:

Figure 1BC, 2BC, 3ABD, 4AB, 5BDEF, 6A-H, S1AB, S2, S3AB, S4A-C

c) Please cite the location of the data clearly in all relevant main and supplementary Figure legends, e.g. “The data underlying this Figure can be found in S1 Data” or “The data underlying this Figure can be found in https://doi.org/10.5281/zenodo.XXXXX”

d) For figures containing FACS data (Figures 5AC), please provide the FCS files and a picture showing the successive plots and gates that were applied to the FCS files to generate the figure. We ask that you please deposit this data in the FlowRepository (https://flowrepository.org/) and provide the accession number/URL of the deposition in the Data Availability Statement in the online submission form.

e) The Ethics statement needs to be the first subheading in the Methods sections the Material & Methods section. You currently have it before “Statistical analysis”; please move it before all subsections.

f) Please ensure that your Data Statement in the submission system accurately describes where your data can be found and is in final format, as it will be published as written there.

g) Per journal policy, if you have generated any custom code during the course of this investigation, please make it available without restrictions upon publication. Please ensure that the code is sufficiently well documented and reusable, and that your Data Statement in the Editorial Manager submission system accurately describes where your code can be found.

We expect to receive your revised manuscript within two weeks.

*Published Peer Review History*

*Press*

Sincerely,

Melissa

Melissa Vazquez Hernandez, Ph.D.

Associate Editor

PLOS Biology

---

## [Editor Report · Decision Letter 3]

Dear Julie,

Thank you for the submission of your revised Research Article "Egress-enhancing mutation reveals the inefficiency of non-enveloped virus cell exit" for publication in PLOS Biology. On behalf of my colleagues and the Academic Editor, Ken Cadwell, I am pleased to say that we can in principle accept your manuscript for publication, provided you address any remaining formatting and reporting issues. These will be detailed in an email you should receive within 2-3 business days from our colleagues in the journal operations team; no action is required from you until then. Please note that we will not be able to formally accept your manuscript and schedule it for publication until you have completed any requested changes.

PRESS

Sincerely, 

Melissa

Melissa Vazquez Hernandez, Ph.D., Ph.D.

Associate Editor

PLOS Biology
